# The rise of South–South trade and its effect on global $CO_2$ emissions

Jing Meng [1,2,3], Zhifu Mi [2,4], Dabo Guan [2,5], Jiashuo Li[6], Shu Tao [3], Yuan Li[2,7], Kuishuang Feng [8], Junfeng Liu[3], Zhu Liu [5,9], Xuejun Wang[3], Qiang Zhang[5] & Steven J. Davis[10]

Economic globalization and concomitant growth in international trade since the late 1990s have profoundly reorganized global production activities and related $CO_2$ emissions. Here we show trade among developing nations (i.e., South–South trade) has more than doubled between 2004 and 2011, which reflects a new phase of globalization. Some production activities are relocating from China and India to other developing countries, particularly raw materials and intermediate goods production in energy-intensive sectors. In turn, the growth of $CO_2$ emissions embodied in Chinese exports has slowed or reversed, while the emissions embodied in exports from less-developed regions such as Vietnam and Bangladesh have surged. Although China's emissions may be peaking, ever more complex supply chains are distributing energy-intensive industries and their $CO_2$ emissions throughout the global South. This trend may seriously undermine international efforts to reduce global emissions that increasingly rely on rallying voluntary contributions of more, smaller, and less-developed nations.

[1] Department of Politics and International Studies, University of Cambridge, Cambridge CB3 9DT, UK. [2] Water Security Research Centre, School of International Development, University of East Anglia, Norwich NR4 7TJ, UK. [3] Laboratory for Earth Surface Processes, College of Urban and Environmental Sciences, Peking University, Beijing 100871, China. [4] Bartlett School of Construction and Project Management, University College London, London WC1E 7HB, UK. [5] Department of Earth System Science, Ministry of Education Key Laboratory for Earth System Modeling, Tsinghua University, Beijing 100084, China. [6] State Key Laboratory of Coal Combustion, Huazhong University of Science and Technology, Wuhan 430074, China. [7] Institute of Resource, Environment and Sustainable Development, Jinan University, Guangzhou 510632, China. [8] Department of Geographical Sciences, University of Maryland, College Park, MD 20742, USA. [9] Tyndall Centre for Climate Change Research, University of East Anglia, Norwich NR4 7TJ, UK. [10] Department of Earth System Science, University of California, Irvine, CA 92697, USA. These authors contributed equally: Jing Meng, Zhifu Mi. Correspondence and requests for materials should be addressed to D.G. (email: dabo.guan@uea.ac.uk) or to J.L. (email: jfliu@pku.edu.cn) or to S.J.D. (email: sjdavis@uci.edu)

International trade increased >50% from 2005 to 2015, with ~60% of the increase tied to rising exports from developing countries[1], which is also known as Global South[2]. Yet over the same period, South–South trade (i.e., among developing countries) has grown even faster—more than tripling—to reach 57% of all developing country exports (US$9.3 trillion) in 2014[3]. The rapid growth in South–South trade reflects a fragmenting of global supply chains whereby early production stages of many industries have relocated from countries like China and India to lower-wage economies[4, 5], a trend that has accelerated since the global financial crisis in 2008[6]. A host of new institutions such as the South–South Cooperation Fund, China's Belt and Road Initiative and the Asian Infrastructure Investment Bank[7–10] have also emerged to support trade and investments among developing nations, indicating that strong growth in South–South trade will continue in the future.

In addition to their important implications for global economic development, these trends will affect the magnitude and regional distribution of future global $CO_2$ emissions[11]. Whereas previous studies have focused on the offshoring of production activities and emissions from developed countries to developing countries[12–14], relatively little attention has been specifically paid to the rapid rise of South–South trade since the 2008–2009 global financial crisis. Yet the period since 2009 has also witnessed decreases in Chinese coal consumption that underpin a leveling off of global $CO_2$ emissions[15], as well as the forging of the Paris Agreement whereby nations are determining their contributions to the global effort to reduce $CO_2$ emissions[16]. Given the immense importance of these developments in determining the future global climate, it seems almost equally important to understand how South–South trade may be contributing to changing economic production and structure, and how a diffusion of emissions might affect the efficacy of the Paris Agreement.

To address these questions, here we use the latest released GTAP data to quantify the effects of South–South trade on regional and sectoral $CO_2$ emissions between 2004 and 2011. In summary, we use international trade and $CO_2$ emissions data from 2004, 2007 and 2011 to track emissions related to both intermediate and final goods and services from 57 industry sectors that are traded among 129 regions (101 regions are individual countries)[17]. To facilitate presentation, we aggregate results into ten regions according to the geographical proximity and level of economic development. These include six developing regions of the global South: China, India, Middle East and North Africa, Latin America and the Caribbean, Sub-Saharan Africa, and Other Asia and Pacific; and other (mostly developed) regions of U.S.-North America, Western Europe, Eastern Europe and the former Soviet Union, and developed Asia-Pacific regions[12, 18–20]. Our findings show that some production activities are relocating from China and India to other developing regions, particularly raw materials and intermediate goods production in energy-intensive sectors. The trend implies that complex supply chains are distributing energy-intensive industries and their $CO_2$ emissions throughout the global South.

## Results

**Growth in emissions embodied in South–South trade 2004–2011.** In total, $CO_2$ emissions embodied in goods and services exported from developing countries increased by 46% between 2004 and 2011, from 2.2 Gt to 3.3 Gt (Supplementary Table 1). Although a substantial and growing quantity of these emissions were embodied in exports to developed regions (1.8 Gt in 2004 and 2.2 Gt in 2011, growing by an average of 2.9% per year), the emissions embodied in South–South trade increased much more rapidly: from 0.47 Gt in 2004 to 1.1 Gt in 2011 (1.33%

per year). The growth is mainly driven by the increasing export volume (i.e., exports per capita (0.75 Gt) and population (0.07 Gt)), and partly offset by decline in emission intensity (0.1 Gt).

The shading of regions in Fig. 1 indicates the magnitude of net imports (reds) or net exports (blue) embodied in South–South trade in each year 2004, 2007, and 2011. Arrows in the figure represent the ten largest South–South fluxes of embodied emissions in 2004, 2007, and 2011 (Fig. 1a, b, c, respectively). Although many regions in the global South became larger net importers over the period, the magnitude of carbon embodied in exports between the developing regions also increases, including China's exports to all other developing regions, but also (1) Middle East and North Africa's exports to China, (2) India's exports to the Middle East and North Africa, and (3) Other Asia and Pacific's exports to China (Fig. 1 and Supplementary Fig. 1). In other words, emissions embodied in both imports and exports have significantly increased in South–South trade during 2004–2011.

In the period concerned, more than half of the growth in emissions embodied in South–South trade were related to exports from China to other developing regions, which grew by 199% (see Supplementary Fig. 1) or 380 million tonnes (Mt). This amount is comparable to United Kingdom's annual $CO_2$ emission in 2016. China's exports to India accelerated most rapidly, with an average growth rate of 23% per year. The growth of export per capita in China would have increased exported emissions by 0.16 Gt during 2004–2007 and 0.27 Gt during 2007–2011 if other factors

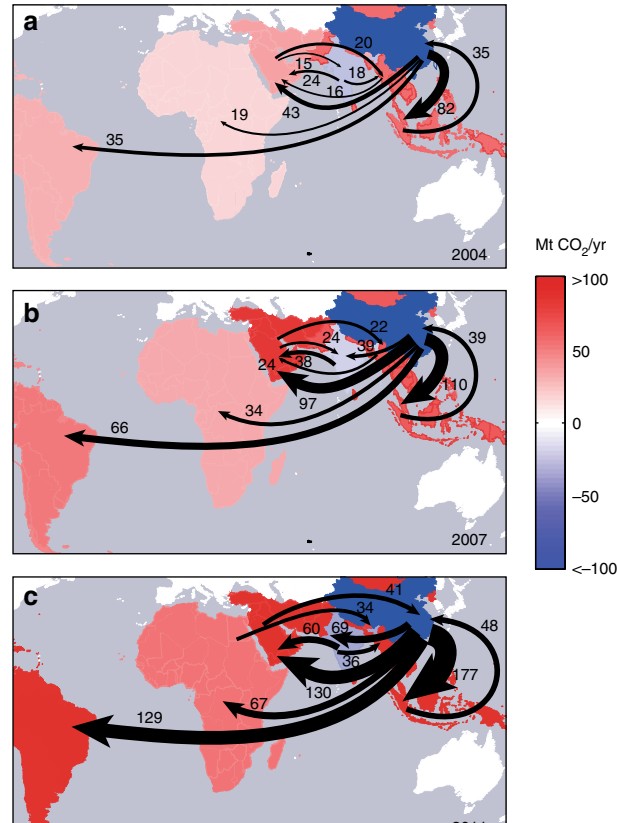

**Fig. 1** Changes in net emissions embodied in South–South trade and largest South–South transfers. Shading indicates regional differences between emissions embodied in imports and exports (i.e., net emissions embodied in trade) with net exporters blue and net importers red. Arrows in each panel show the ten largest South–South transfers of embodied emissions in 2004 (**a**), 2007 (**b**), and 2011 (**c**)

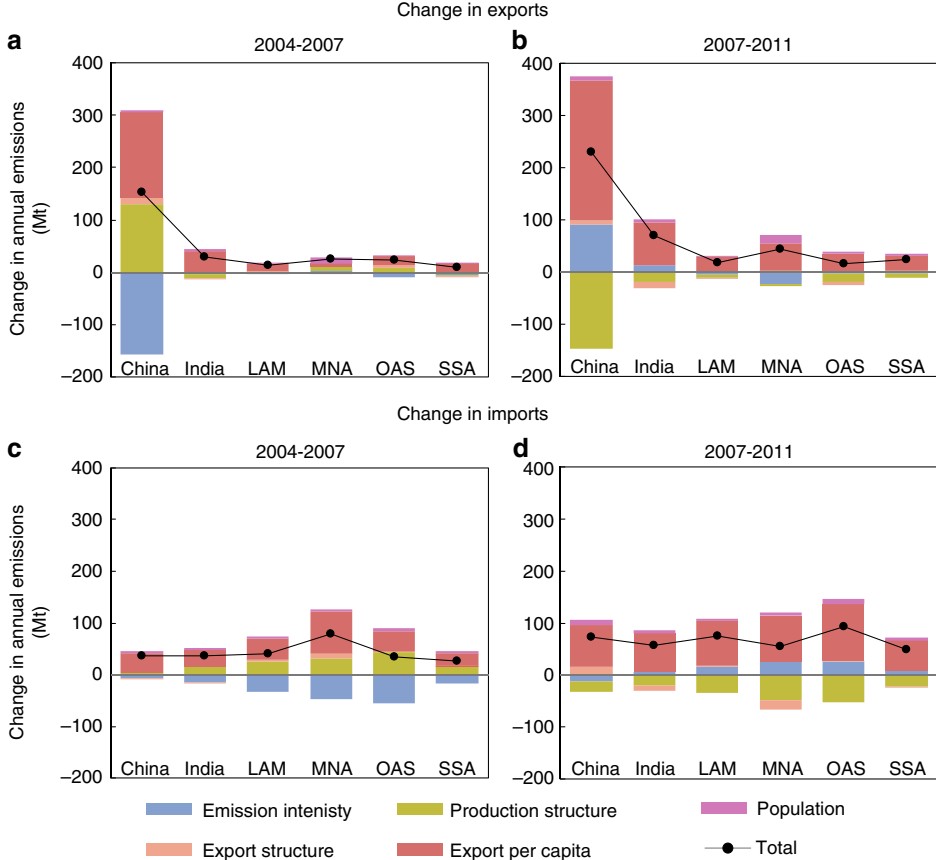

**Fig. 2** Contributions of different factors to changes in emissions embodied in trade. Bars show the contribution of factors to change in exports during 2004–2007 (**a**) and 2007–2011 (**b**), and change in imports during 2004–2007 (**c**) and 2007–2011 (**d**). The developing regions are aggregated into 6 regions, as shown in Supplementary Table 3. Columns show the contributions of different driving factors

were constant (Fig. 2 and Supplementary Fig. 2). The main decelerator was emission intensity from 2004 to 2007, which would have decreased China's export emissions to developing regions by 0.16 Gt (Fig. 2). From 2007 to 2011, production structure change was the strongest factor, offsetting export emissions by 0.15 Gt. Over the same period, emissions embodied in imports to China also increased very rapidly: 137% or 110 Mt that is equivalent to Czech Republic's carbon emissions in 2015. In particular, imports from Other Asia and Pacific rose 39% (14 Mt) and imports from Sub-Saharan Africa increased 249% (45 Mt), mainly due to their increasing imports (i.e., exports from other regions).

**Emissions embodied in China's and India's trade**. As the largest developing countries in the world by both population and economic size, China and India are central to the growing South–South trade. The different comparative advantages of developing countries provide grounds for strong economic exchange. Exports from other developing regions—especially African countries—to China have been predominantly of extractive products, minerals and petroleum. China's resource-intensive growth model—propelled by heavy infrastructure spending and its manufacturing machine—requires a large amount of raw material inputs. Similarly, India imported a range of inputs from other developing regions such as petroleum from Latin America and metals from Sub-Saharan Africa. Both economies export large quantities of manufactured products, and as the world's two fastest-growing economies may serve as the

conduits of global demand for manufactured goods for decades to come.

Figure 3 shows the sectoral breakdown of $CO_2$ emissions embodied in intermediate and final products traded between developing regions and China (Fig. 3a) and India (Fig. 3b) in 2011. Of the 190 Mt $CO_2$ embodied in imports to China from other developing regions, 93% were embodied in intermediate products, and in particular mining products (76 Mt) from Sub-Saharan Africa, India, and Latin America (pale yellow bars in Fig. 3a). China's growing appetite for mined raw materials from Africa (mainly petroleum) led to an increase of 20 Mt $CO_2$ in emissions embodied in China's imports from Sub-Saharan Africa in the period 2004–2011, 83% of the total growth from Sub-Saharan Africa to China (Supplementary Fig. 3). Embodied emissions in China's imports from India increased 22 Mt over 2004–2011, which was mainly sourced from mining and heavy industry sectors (Supplementary Fig. 3). Comparing with trade with Africa, China imported iron ores and other metal ores from India, which was the third largest iron ores provider to China.

Of the 572 Mt $CO_2$ embodied in Chinese exports to other developing regions, 62% were also embodied in intermediate products, but consisting largely of manufactured goods such as machinery (36%). This reflects this importance of China's manufacturing industry to markets in other developing regions. China's exports of machinery and equipment to Sub-Saharan Africa, Latin America and the Caribbean, and the Middle East and North Africa increase (Fig. 3) mainly drives emission-intensive production processes in China, whose role as factory for the world has been confirmed by previous studies[21–23]. Although

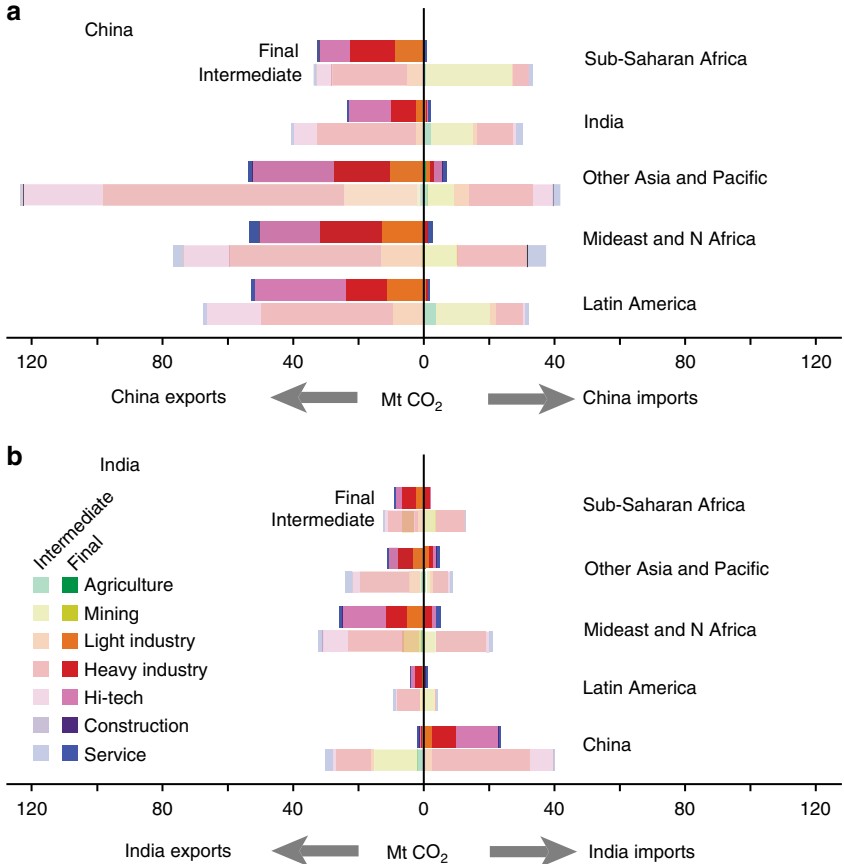

**Fig. 3** Magnitude and composition of emissions embodied in trade between developing regions in 2011. The sectors are the aggregation of 57 sectors and details are shown in Supplementary Table 4. Bars show the balance of $CO_2$ emissions embodied in trade between developing regions and China (**a**) and India (**b**). Colors indicate the sectors of traded products, with final and intermediate products differentiated by both the intensity of shading and separate bars

China produces for the rich developed countries sustained, China's embodied emissions grew largely by producing for the poor developing populations.

Emissions embodied in India's imports from other developing regions also grew three-fold from 44 Mt in 2004 to 135 Mt in 2011. Intermediate products accounted for a large proportion (71%) of the total $CO_2$ emissions embodied in such imports. Heavy industrial products (e.g., chemicals and preliminary processed metal) dominate Indian intermediate imports increase. India imports large amount of final products from China from Hi-tech and heavy industry sectors. Embodied emissions in Indian final products import from China reached 25 Mt in 2011 (Fig. 3b), which has increased by 21 Mt since 2004 (Supplementary Fig. 3).

Of the 165 Mt of India's $CO_2$ emissions embodied in export, 68% were embodied in intermediate products and 32% in final products. Indian intermediate products forms product basis for further processing in other developing countries. Final products have been traded to global South, except China. This implies that India starts to compete with China as a manufacturer for other developing regions, while India required importing higher value added products from China. In particular, the emissions embodied in China's exports to Middle East and North Africa increased by 54 Mt from 2004 to 2007, but dropped to 33 Mt from 2007 to 2011. By contrast, the emissions embodied in India's exports to Middle East and North Africa increased by 15 Mt and 21 Mt during the period 2004–2007 and 2007–2011, respectively. This is mainly because of decline in export volume change's contribution from China (pink and red) and a double increase of that from India (Supplementary Fig. 2).

**China's change as an export platform**. Although China's manufacturing base continues to grow, the rate has tapered in recent years. Because of rising labor costs in China, low-end manufacturers plan to relocate lower-cost foreign economies in the coming years[24]. For example, labor-intensive and resource-intensive production of textiles and apparels has already begun shifting from China to other Asian countries such as Bangladesh and Vietnam[25, 26]. This shift is evidenced by increasing $CO_2$ emissions embodied in textiles exported from other Asia and Pacific regions which rose by an annual rate of 21–22% during 2004–2011 (Fig. 4). In particular, countries like Bangladesh and Vietnam, were able to accumulate their embodied emissions in textiles exports by 175% and 236% between 2004 and 2011 (from 0.61 and 0.86 Mt to 1.68 and 2.89 Mt), respectively. The growth rate of emissions embodied in China's exported textile decreased from an average of 8% per year in the period 2004–2007 to 5% per year in the period 2007–2011 (Fig. 4). By contrast, the emissions embodied in textiles imported to China changed in the opposite trend, decreased at an annual rate of 9% in the period of 2004–2007 but a slight increased 1% per year in the period of 2007–2011 (Supplementary Fig. 4).

Businesses looking for low-cost production are increasingly considering countries other than China, particularly South and Southeast Asian countries such as Indonesia, Vietnam, and Thailand. Emissions embodied in exports from several Chinese industry sectors (e.g., textiles and wearing apparel, ferrous metals) have grown slowly or even decreased since 2004, while emissions embodied in India's, other Asia and Pacific's exports still have an increasing and high growth rate, respectively (see details in Supplementary Data 1). Although foreign direct investment (FDI)

Change in emissions embodied in exports

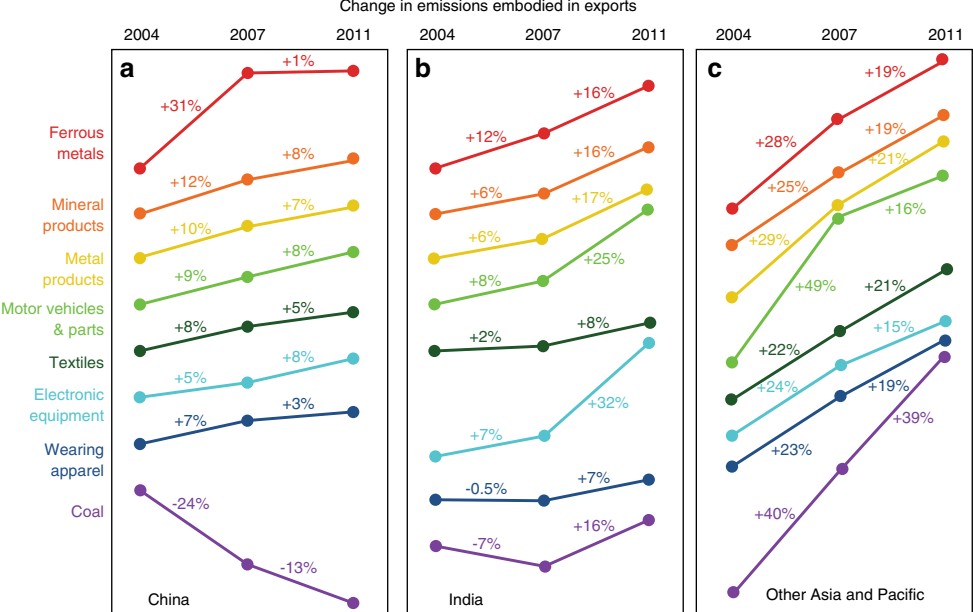

**Fig. 4** Annual changes in emissions embodied in selected types of exported products to the rest of World. Lines show relative changes in the emissions embodied in different types of products exported by China (**a**), India (**b**), and other Asia and Pacific region (**c**). See details of in Supplementary Data 1

flows to China doubled from 2004 and reached a record level of $124 billion in 2011, flows to the services sector surpassed those to manufacturing for the first time[27]. FDI to India grew six-fold from 2004 to 2011 and is expected to increase faster than China. Overall, China continues to be in the top spot as investors' preferred destination for FDI[28], but the rising wages and production costs will enhance the relative competitiveness of South and Southeast Asian countries in manufacturing.

Changes in Chinese manufacturing are being driven by rising labor cost, as well as rising real estate costs and stricter environmental regulations. China is also intentionally transitioning to a more service-based economy, so that tax-related incentives for manufacturing are becoming increasingly difficult to obtain. Meanwhile, nations such as India, Indonesia, Vietnam, Thailand, and others are building critical mass and beginning to attract more businesses. Although China may no longer be the lowest-cost export platform, it will likely continue attracting manufacturing investment to serve domestic consumers and possibly riches from global South and to support a shift toward higher-value manufactured goods[29, 30]. For example, the annual growth rate of emissions embodied in electronic equipment increased from 5% between 2004 and 2007 to 8% between 2007 and 2011.

The carbon intensity of China's economy ($CO_2$ emissions per unit of GDP) decreased by 27% from 2004 to 2015 (Supplementary Table 2), reflecting a shift away from low-value-added manufacturing. Indeed, a 6.5% decrease in Chinese coal consumption over the period 2013–2015 underpins the flattening of global $CO_2$ emissions since 2013[31–33]. However, the carbon intensity of the next phase of global economic development will determine whether ambitious climate targets such as stabilizing at 2 °C will be met, and our findings depict the nascent rise of energy-intensive and emissions-intensive production activities in other Asian countries such as Vietnam and Pakistan. Thus, even as the share of coal in China's energy consumption fell, the share in India and other developing Asian countries increased by 9.3% and 10%, respectively, between 2013 and 2015[34]. The success of the Paris Agreement and international climate mitigation efforts may therefore depend on curtailing growth of coal-based energy and emissions in now-industrializing and urbanizing countries[35].

Otherwise, countries like China and India may meet their nationally determined contribution (NDC) under the Paris Agreement[36, 37] by hollowing out low-value, energy-intensive manufacturing, and offshoring those activities to emerging markets elsewhere in Asia with less stringent climate policy measures. The consumption-induced or weak carbon leakage due to comparative advantage may be extended to policy-induced or strong carbon leakage in response to the different climate policies[38]. Reducing such carbon leakage among developing nations and successfully mitigating climate change thus urgently depends on decarbonizing not only energy systems in developed countries but also the entire process of industrialization[39].

## Methods

**Emissions embodied in trade**. Originally developed by Leontief[40], environmental input-output analyses (EIOs)[40] have been widely used to illustrate the economy-wide environmental repercussions triggered by economic activities. By extending EIOs to multi-regions, emissions embodied in bilateral trade (EEBT) enables to determine the interconnection of sectors in different regions with respect to environmental changes and is suitable for analyzing bilateral relationship[14, 41, 42]. EEBT methods considers the total exports from a country covering all intermediate and final products. Here, we use the global multi-regional input-output tables of 2004, 2007 and 2011 to estimate the emissions from the production of traded products[14, 43]. For each region, the monetary balance is

$$\mathbf{x}^r = \mathbf{Z}^r + \mathbf{y}^r + \sum_s \mathbf{e}^{rs} - \sum_s \mathbf{e}^{sr} \qquad (1)$$

where $\mathbf{x}^r$ is a vector for sectoral total outputs in region $r$; $\mathbf{Z}^r$ represents the domestic and imported industry requirements in region $r$; $\mathbf{y}^r$ is the final demand (household, government and investment) in region $r$ of domestic and imported products; $\mathbf{e}^{rs}$ is the exports from region $r$ to region $s$ ($r \neq s$) and $\mathbf{e}^{sr}$ is the imports in region $r$ from region $s$. In EEBT, imports are removed from $\mathbf{Z}^r$ and $\mathbf{y}^r$ to focus on the domestic production only[44]:

$$\mathbf{x}^r = \mathbf{Z}^{rr} + \mathbf{y}^{rr} + \sum_s \mathbf{e}^{rs} \qquad (2)$$

where imports to $r$ are expressed as

$$\mathbf{m}^r = \sum_s \mathbf{e}^{sr} = \sum_s \mathbf{Z}^{sr} + \sum_s \mathbf{y}^{sr} \qquad (3)$$

The total direct and indirect domestic emissions to produce a unit of final consumption is

$$\mathbf{h}^r = \mathbf{F}^r (\mathbf{I} - \mathbf{A}^{rr})^{-1} \qquad (4)$$

where $\mathbf{F}^r$ is the direct emission intensity in region $r$, which is calculated by each sector's $CO_2$ emissions divided by the sector's total output $x$[45]. The $CO_2$ emissions used in this study rely on data from the GTAP which reflects the use of energy and fossil fuels by each region sector. The emissions are reduced by 9% for the agriculture and industry sectors according to our previous study[46]. To extend the coverage of emission data, we also include emissions from cement production, which is attributed to sector of Non-Metallic Minerals[47]. $\mathbf{L} = (\mathbf{I} - \mathbf{A})^{-1}$ is the Leontief inverse matrix, which captures both direct and indirect inputs to satisfy one unit of final demand in monetary value, $(\mathbf{I} - \mathbf{A}^{rr})^{-1}$ only consider the domestic supply chain in region $r$.

The total direct and indirect emissions in region $r$ to produce the products which are exported to region $s$ are

$$T^{rs} = \mathbf{F}^r (\mathbf{I} - \mathbf{A}^{rr})^{-1} \mathbf{e}^{rs} \tag{5}$$

**Structural decomposition analysis.** Index decomposition analyses (IDAs) and structural decomposition analyses (SDAs) are two decomposition methods that are widely used to quantify the driving factors of a dependent variable, such as energy consumption or $CO_2$ emissions. SDAs enable us to distinguish a range of production effects and final consumption effects that IDAs fail to capture[48–50], and they are capable of assessing both direct and indirect effects along the entire supply chain[51]. The typical SDA can quantify the degree of change in emission transfers among regions when only the trade structure increases and all other factors remain the same[51]. SDAs can decompose the bilateral emission transfers into changes in constituent parts. In our analysis, these constituent parts are emission intensity, production structure, export structure, export per capita and population effect[12, 19, 52, 53].

Previous studies have applied a similar SDA approach to determine the underlying factors affecting increases in China's consumption-based $CO_2$ emissions[12] or China's emissions from production of exported-product[52]. Clearly, emissions embodied in trade have played an increasingly important role in regional $CO_2$ emissions in a globalizing world; however, insufficient attention has been paid to the driving forces of bilateral emission transfers, especially the developing regions, which are the focuses of our study.

The total emissions in region r to produce the products which are exported to region s can be expressed as follows:

$$\begin{aligned} T^{rs} &= \mathbf{F}^r (\mathbf{I} - \mathbf{A}^{rr})^{-1} \mathbf{e}^{rs} \\ &= \sum_i \sum_j f_i^r L_{ij}^{rr} \frac{e_j^{rs}}{e^{rs}} \frac{e^{rs}}{P^r} P^r \\ &= \sum_i \sum_j f_i^r L_{ij}^{rr} S_j^{rs} M^{rs} P^r \end{aligned} \tag{6}$$

where $P^r$ and $M^{rs}$ can reflect the change in export volume, which is population in region $r$ and export volume per capita from region $r$ to region $s$, respectively; $e_j^{rs}$ is the share of the export of products in sector $j$ from region $r$ to region $s$ in the total exports from region $r$ to region $s$; $L_{ij}^{rr}$ indicates the total inputs from sector $i$ to produce one unit of output in sector $j$ in region $r$,; and $f_i^r$ is the emissions for a unit of output in sector $i$ in region $r$.

Thus, the change in the bilateral emission transfers between two points in time (indicated by the subscripts 0 and 1) can be expressed as $\Delta T^{rs} = T_1^{rs} - T_0^{rs}$. However, a unique solution for the decomposition is not available[18, 21, 54, 55]. For the case of m factors, the number of possible complete decompositions without any residual terms is equal to $m!$[18]. Because of large numbers of sectors and regions in this study, we follow the methods of previous studies and use the average of the so-called polar decompositions as an approximation of the average of all $m!$ decompositions[12, 18]. The two polar decompositions ($\Delta T_a^{rs}$ and $\Delta T_b^{rs}$) are as follows:

$$\begin{aligned} \Delta T_a^{rs} &= \sum_i \sum_j (\Delta f_i^r) L_{ij1}^{rr} S_{j1}^{rs} M_1^{rs} P_1^r + \sum_i \sum_j f_{i0}^r (\Delta L_{ij}^{rr}) S_{j1}^{rs} M_1^{rs} P_1^r \\ &+ \sum_i \sum_j f_{i0}^r L_{ij0}^{rr} (\Delta S_j^{rs}) M_1^{rs} P_1^r + \sum_i \sum_j f_{i0}^r L_{ij0}^{rr} S_{j0}^{rs} (\Delta M^{rs}) P_1^r \\ &+ \sum_i \sum_j f_{i0}^r L_{ij0}^{rr} S_{j0}^{rs} M_0^{rs} (\Delta P^r) \\ &= \Delta f_a + \Delta L_a + \Delta S_a + \Delta M_a + \Delta P_a \end{aligned} \tag{7}$$

$$\begin{aligned} \Delta T_b^{rs} &= \sum_i \sum_j (\Delta f_i^r) L_{ij0}^{rr} S_{j0}^{rs} M_0^{rs} P_0^r + \sum_i \sum_j f_{i1}^r (\Delta L_{ij}^{rr}) S_{j0}^{rs} M_0^{rs} P_0^r \\ &+ \sum_i \sum_j f_{i1}^r L_{ij1}^{rr} (\Delta S_j^{rs}) M_0^{rs} P_0^r + \sum_i \sum_j f_{i1}^r L_{ij1}^{rr} S_{j1}^{rs} (\Delta M^{rs}) P_0^r \\ &+ \sum_i \sum_j f_{i1}^r L_{ij1}^{rr} S_{j1}^{rs} M_1^{rs} (\Delta P^r) \\ &= \Delta f_b + \Delta L_b + \Delta S_b + \Delta M_b + \Delta P_b \end{aligned} \tag{8}$$

The average of the polar decomposition is determined as follows[18]:

$$\begin{aligned} \Delta T^{rs} &= \tfrac{1}{2} (\Delta T_a^{rs} + \Delta T_b^{rs}) \\ &= \tfrac{1}{2} (\Delta f_a + \Delta f_b) + \tfrac{1}{2} (\Delta L_a + \Delta L_b) + \tfrac{1}{2} (\Delta S_a + \Delta S_b) \\ &+ \tfrac{1}{2} (\Delta M_a + \Delta M_b) + \tfrac{1}{2} (\Delta P_a + \Delta P_b) \\ &= \Delta f + \Delta L + \Delta S + \Delta M + \Delta P \end{aligned} \tag{9}$$

where $\Delta T^{rs}$ is the growth in bilateral emission transfers between two points in time, which in this study was from 2004 to 2007 and from 2007 to 2011; and $\Delta f$, $\Delta L$, $\Delta S_a$, $\Delta M$, and $\Delta P$ refer to the emission intensity effect, production structure effect, export structure effect, export per capita, and population effect, respectively.

**Data sources.** The economic input-output data, population, energy consumption and $CO_2$ emissions of each sector are all based on version 9 of the Global Trade Analysis Project (GTAP) database[56]. These include six developing regions of the global South: China, India, Middle East and North Africa, Latin America and the Caribbean, Sub-Saharan Africa, and Other Asia and Pacific; and other (mostly developed) regions of U.S.-North America, Western Europe, Eastern Europe and the former Soviet Union, and developed Asia-Pacific regions (Supplementary Table 1 and Data 2)[12, 18–20].

The economic data from the GTAP database are in current prices (US dollars), and to remove the impact of inflation on the monetary output, we use the appropriate producer price index (PPI) to adjust all of the monetary data to provide a consistent analysis from 2004 to 2011. The PPI can be derived from pricing data for seven categories published by National Account Main Aggregates Database[57], which are mapped to 57 sectors of global Multi-regional Input-output table (Supplementary Table 5). Emissions from fossil fuel combustion by each region are from GTAP database[56], and emission from cement production is from Andrew[47]. We are aware that China's $CO_2$ emissions are derived from various sources; thus, we lowered the fossil fuel $CO_2$ emissions from agriculture and industry in China by 9% according to our previous studies[46, 58]. Our analysis is global and includes 129 regions (Supplementary Data 2), and the detailed results are aggregated into 10 regions based on geography and economic levels (Supplementary Table 3) for ease of understanding.

**Data availability.** All the original data can be obtained from given data sources. Supplementary Data 1 is annual changes in emissions embodied in exports in selected regions. Supplementary Data 2 is the definition of regions. Supplementary Data 3 is South–South transfers of embodied emissions in 2004, 2007, and 2011. All datasets generated during this study are available from the corresponding author on reasonable request.

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

## Acknowledgements

This work was supported by the National Key R&D Program of China (2016YFA0602604) and National Natural Science Foundation of China (41629501, 71761137001, 71704060, 41571130010, 41671491, and 41390240), National Social Science Foundation of China (15ZDA054), National Key Research and Development Program of China 2016YFC0206202, the 111 Project (B14001), the UK Economic and Social Research Council (ES/L016028/1) Natural Environment Research Council (NE/N00714X/1 and NE/P019900/1) and British Academy Grant (AF150310) and the Philip Leverhulme Prize.

## Author contributions

J.M. and D.G. designed the study. J.M. performed the analysis and prepared the manuscript. J.M., Z.M., D.G., and S.J.D. interpreted data. D.G. coordinated and supervised the project. All authors (J.M., Z.M., D.G., J.-S.L., S.T., Y.L., K.F., J.-F.L., Z.L., X.W., Q. Z., and S.J.D.) participated in the writing of the manuscript.

## Additional information

**Competing interests:** The authors declare no competing interests.

