## [Peer Review File · Nature Communications]

Reviewers' comments:

Reviewer #1 (Remarks to the Author):

The authors investigated an interesting topic on the impacts of south-south trade on the embodied emission flows among China and other countries in the world. They chose using the GTAP database in the analysis for 2004, 2007 and 2011 to capture the changes before and after the global financial crisis. Although the topic is new, the results presented are not surprising comparing with the recent studies reported by the same authors using other multi-region datasets, including Jiang and Guan (2016; *Applied Energy* 184, 1132-1141; "Determinants of global CO2 emissions growth") used the WIOD database for 1995-2009 and Jiang and Guan (2017; *Energy Policy* 109, 734-746; "The global CO2 emissions growth after international crisis and the role of international trade") used the OECD-ICIO database for 2008-2011. Besides, the OECD also publishes regular updates on the detailed embodied emission flows for global countries covered in the OECD-ICIO database for time series year 1995-2011 (https://stats.oecd.org/Index.aspx?DataSetCode=IO_GHG_2015).

To capture the changes before and after the global crisis, the authors only selected three years in the analysis, which is not sufficient. Similar work was done by Peters et al. (2011; *PANS* 108, 8903-8908; "Growth in emission transfers via international trade from 1990 to 2008") who also used the GTAP database but gave the time-series estimates of embodied emission flows for global countries in period 1990-2008. The WIOD and OECD-ICIO database with time-series data might be more suitable for the analysis required in this paper. Driving forces to the results happened in any two years can be further explored by the Structural Decomposition Analysis like Jiang and Guan (2017).

Reviewer #3 (Remarks to the Author):

This article focuses on the effect of the increased South-South trade on global CO2 emissions. Changes in the emissions embodied in the South-South trade between 2004 and 2011 are analyzed using the environmental input-output methods. The article finds a significant increase in the emissions embodied in the trade between developing countries, which has been overlooked. The findings are of interest and will have potential influences in the field of climate change. That said, I have the following comments which might help strength the article.

1. On analysis and discussions

1.1 Weak policy implications. The article provides rich analysis on the changes in the emissions in South-South trade, showing an overview of the changes, the changes related to China and India, and changes in China's exports specifically. From my point of view, the novelty of this article is reflected on its potential influences on climate policy, which is, however, not well-discussed in the article. I think a more in-depth discussion on the potential influences of the findings, especially on the climate policy, could help make the article not only more interesting, but also valuable. Therefore, I would expect a comprehensive and thorough discussion about the findings of the article, which could give a full view of all the changes, talk about the interesting points (e.g. the start of India's competition as a manufacturer with China, and China's changes as an export platform), and focus on the potentially influences of the findings under the circumstance of the Paris Agreement and the current changes in globalization.

1.2 References or data to support the argument of US companies' choice of moving to Mexico instead of China. When talking about China's changes as an export platform, the article argues that increase in the emissions in moto vehicles and transportation equipment from Mexico is "an early reflection" of the fact that "a growing number of US companies have moving production to

Mexico" (line 179~182). This looks a bit arbitrary since the report referred here¹ is only about the period 2007-2011. I would expect to see more evidence, such as data of foreign investment, for the earlier period 2004-2007.

1.3 Discussions on potential to undermine the Paris Agreement. When talking about China's carbon intensity, the article shows worries about the potential of the increased South-South trade to undermine the effect of the Paris Agreement (line 205~208), which is reasonable. However, since the Paris Agreement is a 'bottom-up' approach where every country has its nationally determined contribution (NDC), the argument in the article is weakened. Therefore, more discussions should be involved on this issue to make the argument stronger.

2. On methods and data

2.1 Region aggregation and classification. This article aggregates the countries/regions into 10 regions "according to geographical proximity and level of economic development (line 80~81)", of which the Global South is the focus of this article and is equal to 'developing countries' in the article. Specifically, the 'Four Asian Tigers' (Republic of Korea; Taiwan, China; Hong Kong, China; and Singapore) are divided into "Other Asia and Pacific" and seen as developing countries/regions, which does not seem to be appropriate because these four countries/regions are well-developed in terms of economy. And what further confused me is that from Figure 1, we could see that the Republic of Korea is actually not included in the analysis (not shaded).

Since the region classification adopted by the authors (the version described in the text) is consistent with that of the Kyoto Protocol (Annex B and non-Annex B), it is reasonable to use it in the field of climate change. Therefore, I would suggest that a consistent classification is used throughout the article, and that the bound of the analysis could be shown clearer (for example, add a column for whether a region is developed or developing in Table S2). Moreover, when necessary, explain the reason why the 'Four Asian Tigers' are classified to the developing countries.

2.2 Use of the producer price index (PPI). This article uses the PPI to convert the monetary data in current price into constant price (line 347~349). As the PPI is mostly for commodity goods and perhaps only several services, I would wonder which price indices are used for the services sectors.

2.3 The coverage of the emission data. The CO₂ emission data used in this study are taken from the GTAP 9 database, but the coverage of emission data are not explained in the article. If I did not misunderstand the documentation of the GTAP 9 (<https://www.gtap.agecon.purdue.edu/resources/download/7637.pdf>), their emission data only include those from the fossil fuel combustion. As the production process of cements is also a source of CO₂ emissions (5% in 20112, and can be higher in the developing countries, e.g. around 7% in China in 20113), it might be necessary to discuss the potential uncertainties in the results caused by the ignorance of emissions from cement.

3. Line-by-line comments

Line 59. "The Belt and Road Initiative" is better than "One Belt One Road strategy".

Line 74. 'The latest released GTAP data' would be better.

Line 92. Please check if the number 0.49 is correct, as it is different from that in Table S3 (0.4).

Line 126. 'Of the 190 Mt...', and check the rest of the article.

Line 309. Please clarify in the caption where it is emissions in the exports to the rest of the developing countries or to the rest of world.

Line 344. The last paragraph of the Materials and Methods section looks like an explanation for the data use of the article, yet it seems that a subtitle has been missed for this paragraph. If nothing missed, there is no need to have the subtitle "Emissions embodied in trade".

References

1. Selko A, Vinas T. Nearshoring fuels Mexican manufacturing growth. Industry Week 2012.
2. Le Quéré C, Andres RJ, Boden T, Conway T, Houghton RA, House JI, et al. The global carbon budget 1959–2011. Earth System Science Data Discussions 2012, 5(2): 1107-1157.
3. CEADs. China Emission Accounts and Datasets. <http://www.ceads.net/>; 2017.

Reviewer #4 (Remarks to the Author):

This is a really nice paper. It's simple, gets straight to the point, and is well executed and well written.

I recommend it to be Accepted.

I had a few comments the authors may wish to address before it's published.

In the Abstract or Introduction can you define "south". I think you are using it just to mean trade not involving, US, Canada, and the EU-28. Is that right? (Yes I realize the list of countries is provided in the SI, but this is not so convenient for the reader.)

Line 79, "most regions". Can you be specific: how many are countries, how many are regions? This can really affect results for South America, Africa, and SE Asia.

Line 161: "plans to" should be "plan to"

Line 164: "shift is reflected" bit awkward; consider rephrasing the sentence.

Line 180: "have moving" should be "have moved"

Line 86: Maybe explain here (in the paper, not just in Methods) why you chose GTAP with only 3 years, rather than WIOD, EXIOBASE, or Eora, which all offer continuous time series greater than the period you currently study.

Reviewer #1 (Remarks to the Author):

The authors investigated an interesting topic on the impacts of south-south trade on the embodied emission flows among China and other countries in the world. They chose using the GTAP database in the analysis for 2004, 2007 and 2011 to capture the changes before and after the global financial crisis. Although the topic is new, the results presented are not surprising comparing with the recent studies reported by the same authors using other multi-region datasets, including Jiang and Guan (2016; Applied Energy 184, 1132-1141; “Determinants of global CO₂ emissions growth”) used the WIOD database for 1995-2009 and Jiang and Guan (2017; Energy Policy 109, 734-746; “The global CO₂ emissions growth after international crisis and the role of international trade”) used the OECD-ICIO database for 2008-2011. Besides, the OECD also publishes regular updates on the detailed embodied emission flows for global countries covered in the OECD-ICIO database for time series year 1995-2011 (https://stats.oecd.org/Index.aspx?DataSetCode=IO_GHG_2015).

Response:

This paper presents results of new global trade analysis (with latest available datasets) that reveal a new trend: the rise in CO₂ emissions embodied in exports quickly accumulated contributing by trade between the global South. Moreover, the paper for the first time quantitatively explores the reasons behind such findings.

In particular, this paper shows that this “South-South” trade more than doubled from 2004 to 2011, growing at an annual rate of almost 12%, and that the goods being traded are dominated by raw materials and intermediate goods from energy- and emissions-intensive industry sectors. This paper shows a new phase of globalization, China and India may seek to meet their emissions targets by increasingly outsourcing energy- and carbon emission intensive production to other developing countries such as Vietnam and Bangladesh which have weak or no commitments under the Paris climate agreement. These findings provide insights for the potentially influences under the circumstance of the Paris agreement and the current changes in globalization.

The two papers (Jiang and Guan, 2016, 2017) were crafted by using different models and datasets and discussing emissions trade between developed and developing countries. The results reported by OECD Working Paper (Wiebe and Yamano 2016) lacks details on emissions embodied in traded products and services. This paper focuses on the storyline of rising global south trade and discuss the opportunities in emission mitigation among global south countries. The model developed and results produced are significantly different from previous contributions. Given the broad implications of this South-South trend for global

emissions and climate change mitigation, we believe this paper is extremely salient and will be of interest to a wide range of scientists, policymakers, and the public.

To capture the changes before and after the global crisis, the authors only selected three years in the analysis, which is not sufficient. Similar work was done by Peters et al. (2011; PANS 108, 8903-8908; "Growth in emission transfers via international trade from 1990 to 2008") who also used the GTAP database but gave the time-series estimates of embodied emission flows for global countries in period 1990-2008. The WIOD and OECD-ICIO database with time-series data might be more suitable for the analysis required in this paper.

Response:

This study focuses on the changes in emissions flows between developing regions, while many developing regions have been aggregated to "Rest of the World" in WIOD and OECD-ICIO database, which is not suitable for this study. For example, the latest WIOD database covers forty-three countries, including 7 developing regions, while OECD-ICIO database covers 28 non-OECD regions, which means a large quantity of detailed/specific information on many South-South regions is missing in WIOD and OECD-ICIO database. By contrast, GTAP database covers 77 developing regions, as listed in Table S1 and S2 in supplementary information, enabling the researchers to draw a holistic picture with high resolution.

By following the reviewer's suggestion, we have newly added the structural decomposition analysis (as suggested below) to quantify the contribution of different socioeconomic drivers underlying the rising CO₂ emissions embodied in South-South trade. As this research only pays attention to the impact of global financial analysis in 2008, the periods before financial crisis (2004-2007) and after financial crisis (2007-2011) in GTAP is enough for this study.

Driving forces to the results happened in any two years can be further explored by the Structural Decomposition Analysis like Jiang and Guan (2017).

Response:

We have followed the suggestion and added structural decomposition analysis to quantify the contributions of different socioeconomic drivers (i.e., emission intensity, production structure, export structure, export volume per capita and population) underlying the rising CO₂ emissions embodied in South-South trade.

The revised manuscript includes description of the decomposition method and data (Lines 366-405), updated results and discussion (Lines 90-92, 107-116, 162-167) and new figures (Fig.2, Fig.S3 in the supplementary information).

Reviewer #3 (Remarks to the Author):

This article focuses on the effect of the increased South-South trade on global CO₂ emissions. Changes in the emissions embodied in the South-South trade between 2004 and 2011 are analyzed using the environmental input-output methods. The

article finds a significant increase in the emissions embodied in the trade between developing countries, which has been overlooked. The findings are of interest and will have potential influences in the field of climate change. That said, I have the following comments which might help strength the article.

Response:

Thank you for your constructive comments. This manuscript will be improved substantially by addressing the comments.

1. On analysis and discussions

1.1 Weak policy implications. The article provides rich analysis on the changes in the emissions in South-South trade, showing an overview of the changes, the changes related to China and India, and changes in China's exports specifically. From my point of view, the novelty of this article is reflected on its potential influences on climate policy, which is, however, not well-discussed in the article. I think a more in-depth discussion on the potential influences of the findings, especially on the climate policy, could help make the article not only more interesting, but also valuable. Therefore, I would expect a comprehensive and thorough discussion about the findings of the article, which could give a full view of all the changes, talk about the interesting points (e.g. the start of India's competition as a manufacturer with China, and China's changes as an export platform), and focus on the potentially influences of the findings under the circumstance of the Paris agreement and the current changes in globalization.

Response:

We have improved the discussion by highlighting the points:

1) India's competition as a manufacturer with China and the driving forces (Lines 160-167);

2) China's change as an export platform by comparing: 1, emissions embodied in exports in several sectors in China, India and Southeast Asian countries (Lines 169-182); 2, growth rate of foreign direct investment in China and India in the study period (Lines 188-194);

3) The differentiated responsibilities of developed countries and developing countries in Paris agreement may lead to "strong" carbon leakage in response to climate policy (Lines 214-226).

1.2 References or data to support the argument of US companies' choice of moving to Mexico instead of China. When talking about China's changes as an export platform, the article argues that increase in the emissions in moto vehicles and transportation equipment from Mexico is "an early reflection" of the fact that "a growing number of US companies have moving production to Mexico" (line 179~182). This looks a bit arbitrary since the report referred here is only about the

period 2007-2011. I would expect to see more evidence, such as data of foreign investment, for the earlier period 2004-2007.

Response:

Thank you for your suggestion. We have rephrased this paragraph by replacing the Mexico example with a comparison of FDI growth to China and India. We believe these changes can better reflect the points we are making here (lines 183 – 194).

1.3 Discussions on potential to undermine the Paris Agreement. When talking about China's carbon intensity, the article shows worries about the potential of the increased South-South trade to undermine the effect of the Paris Agreement (line 205~208), which is reasonable. However, since the Paris Agreement is a 'bottom-up' approach where every country has its nationally determined contribution (NDC), the argument in the article is weakened. Therefore, more discussions should be involved on this issue to make the argument stronger.

Response:

We agree with the reviewer's point that the non-binding nature of the Paris agreement means that China (and other countries) may back away from their stated commitments. However, we have expanded discussion (Lines 214-226) to better explain the possibility that South-South trade could undermine global climate targets without jeopardizing individual country's commitments. For example, the main point in India's NDC is "To reduce the emissions intensity of its GDP by 33 to 35 percent by 2030 from 2005 level" without an absolute emission reduction target. That means India's emissions may still increase with the growing exports. Similarly, China could meet its nationally determined contribution (NDC) under the Paris Agreement by "hollowing out" low-value, energy-intensive manufacturing and offshoring those activities to emerging markets elsewhere in Asia with less stringent climate policy measures.

2. On methods and data

2.1 Region aggregation and classification. This article aggregates the countries/regions into 10 regions "according to geographical proximity and level of economic development (line 80~81)", of which the Global South is the focus of this article and is equal to 'developing countries' in the article. Specifically, the 'Four Asian Tigers' (Republic of Korea; Taiwan, China; Hong Kong, China; and Singapore) are divided into "Other Asia and Pacific" and seen as developing countries/regions, which does not seem to be appropriate because these four countries/regions are well-developed in terms of economy. And what further confused me is that from Figure 1, we could see that the Republic of Korea is actually not included in the analysis (not shaded).

Since the region classification adopted by the authors (the version described in the text) is consistent with that of the Kyoto Protocol (Annex B and non-Annex B), it is reasonable to use it in the field of climate change. Therefore, I would suggest that a

consistent classification is used throughout the article, and that the bound of the analysis could be shown clearer (for example, add a column for whether a region is developed or developing in Table S2). Moreover, when necessary, explain the reason why the 'Four Asian Tigers' are classified to the developing countries.

Response:

Thank you for your suggestion. We have added a column in Table S2 which indicates developing regions in this study. We revise the aggregation of the countries/regions in this study and define a category as "*Developed Asia-Pacific regions*" which includes the 'Four Asian Tigers' (Republic of Korea; Taiwan, China; Hong Kong, China; and Singapore) and Japan, Australia and New Zealand. Detailed results can be found in revised manuscript.

2.2 Use of the producer price index (PPI). This article uses the PPI to convert the monetary data in current price into constant price (line 347~349). As the PPI is mostly for commodity goods and perhaps only several services, I would wonder which price indices are used for the services sectors.

Response:

The National Account Main Aggregates Database provides pricing data for seven categories, which are mapped to 57 sectors of global MRIO (Table S7). Three of the them are related to service sectors, i.e., Wholesale, retail trade, restaurants and hotels (ISIC G-H), Transport, storage and communication (ISIC I) and Other Activities (ISIC J-P).

We have added detailed description of PPI used to convert the monetary data in current price into constant price (Lines 410-416).

2.3 The coverage of the emission data. The CO₂ emission data used in this study are taken from the GTAP 9 database, but the coverage of emission data are not explained in the article. If I did not misunderstand the documentation of the GTAP 9 (<https://www.gtap.agecon.purdue.edu/resources/download/7637.pdf>), their emission data only include those from the fossil fuel combustion. As the production process of cements is also a source of CO₂ emissions (5% in 2012, and can be higher in the developing countries, e.g. around 7% in China in 2013), it might be necessary to discuss the potential uncertainties in the results caused by the ignorance of emissions from cement.

Response:

To reduce the uncertainties in the results, we have included emissions from cement in the revised manuscript. We have added data description (Lines 355-359, Lines 414-416) and updated all the results and figures.

3. Line-by-line comments

Line 59. “The Belt and Road Initiative” is better than “One Belt One Road strategy”.

Response:

Thank you for your suggestion.

Line 74. ‘The latest released GTAP data’ would be better.

Response:

Thank you for your suggestion.

Line 92. Please check if the number 0.49 is correct, as it is different from that in Table S3 (0.4).

Response:

We improve the calculation and analysis by adding the CO₂ emissions from cement production and changing the region aggregation (as mentioned above), so the results have been changed slightly. The new result for CO₂ emissions embodied in trade between developing regions are 0.47. We have thoroughly checked the all numbers in the manuscript to keep consistence and accuracy.

Line 126. ‘Of the 190 Mt...’, and check the rest of the article.

Response:

Thank you for your suggestion, we have changed “of 190 Mt...” to “of the 190 Mt”. We have thoroughly checked the manuscript and changed all similar expressions.

Line 309. Please clarify in the caption where it is emissions in the exports to the rest of the developing countries or to the rest of world.

Response:

We have clarified in the caption of Figure 3 (Figure 4 in revised manuscript) and Figure S3 (Figure S4 in revised SI) that it is emissions in the exports/imports from/to the rest of world.

Line 344. The last paragraph of the Materials and Methods section looks like an explanation for the data use of the article, yet it seems that a subtitle has been missed for this paragraph. If nothing missed, there is no need to have the subtitle “Emissions embodied in trade”.

Response:

Thank you for your suggestion. We have added a subtitle “Data sources” for the last paragraph.

References

1. Selko A, Vinas T. Nearshoring fuels Mexican manufacturing growth. Industry Week 2012.
2. Le Quéré C, Andres RJ, Boden T, Conway T, Houghton RA, House JI, et al. The global carbon budget 1959–2011. Earth System Science Data Discussions 2012, 5(2): 1107-1157.
3. CEADs. China Emission Accounts and Datasets. <http://www.ceads.net/>; 2017.

Reviewer #4 (Remarks to the Author):

This is a really nice paper. It's simple, gets straight to the point, and is well executed and well written.

I recommend it to be Accepted.

I had a few comments the authors may wish to address before it's published.

Response:

Thank you for your positive comment. This manuscript will be improved substantially by addressing the comments.

In the Abstract or Introduction can you define "south". I think you are using it just to mean trade not involving, US, Canada, and the EU-28. Is that right? (Yes I realize the list of countries is provided in the SI, but this is not so convenient for the reader.)

Response:

Thank you for your suggestion. We have added definition of "south" in the Introduction (Lines 48-50).

Line 79, "most regions". Can you be specific: how many are countries, how many are regions? This can really affect results for South America, Africa, and SE Asia.

Response:

Thank you for the suggestion, we have clarified the number of individual countries in the manuscript as below:

"from 57 industry sectors that were traded among 129 regions (101 regions are individual countries, see Table S1)"

Line 161: "plans to" should be "plan to"

Response:

Thank you. Typos are corrected.

Line 164: "shift is reflected" bit awkward; consider rephrasing the sentence.

Response: *"The shift is reflected by..."* has been changed to *"The shift is evidenced by..."*.

Line 180: “have moving” should be “have moved”

Response:

Thank you. Typos are corrected.

Line 86: Maybe explain here (in the paper, not just in Methods) why you chose GTAP with only 3 years, rather than WIOD, EXIOBASE, or Eora, which all offer continuous time series greater than the period you currently study.

Response

We choose GTAP database for two reasons:

First of all, this study focuses on the changes in emissions flows between developing regions, while many developing regions have been aggregated to “Rest of the World” in WIOD and OECD-ICIO database, which is not suitable for this study. For example, the latest WIOD database covers forty-three countries, including 7 developing regions, while latest EXIOBASE database covers 8 developing regions. By contrast, GTAP database covers 77 developing regions, as listed in Table S1 and S2 in supplementary information. Eora database has a heterogeneous classification, which impeding comparing results between countries. The harmonized version has 26-sectors, which is much less than the GTAP database. Sector aggregation has great impact on MRIO uncertainty.

Secondly, we have added structural decomposition analysis to quantify the contribution of different socioeconomic drivers underlying the rising CO₂ emissions embodied in South-South trade. The three years in GTAP which covers both the period before financial crisis (2004-2007) and after financial crisis (2007-2011) is enough for this study.

REVIEWERS' COMMENTS:

Reviewer #1 (Remarks to the Author):

The authors have addressed my major concerns in the revised paper, and included the structural decomposition analysis to show the driving forces to the embodied emission changes in South-South trade. I don't have further comments to add and would recommend it for publication in this journal.

Reviewer #3 (Remarks to the Author):

The authors have made a thorough revision to the manuscript, and I am pleased with the changes.

Reviewer #4 (Remarks to the Author):

I had initially recommended the paper to be Accepted. The revised version is only improved so I maintain my suggestion. The authors have done a suitable job of responding to the referee comments.

REVIEWERS' COMMENTS:

Reviewer #1 (Remarks to the Author):

The authors have addressed my major concerns in the revised paper, and included the structural decomposition analysis to show the driving forces to the embodied emission changes in South-South trade. I don't have further comments to add and would recommend it for publication in this journal.

Response:

Thank you for your recommendation.

Reviewer #3 (Remarks to the Author):

The authors have made a thorough revision to the manuscript, and I am pleased with the changes.

Response:

Thank you for your positive comments.

Reviewer #4 (Remarks to the Author):

I had initially recommended the paper to be Accepted. The revised version is only improved so I maintain my suggestion. The authors have done a suitable job of responding to the referee comments.

Response:

Thank you for your recommendation.